# Biogenic Synthesis of Copper-Based Nanomaterials Using Plant Extracts and Their Applications: Current and Future Directions

**DOI:** 10.3390/nano12193312

**Published:** 2022-09-23

**Authors:** Jei Vincent, Kam Sheng Lau, Yang Chia-Yan Evyan, Siew Xian Chin, Mika Sillanpää, Chin Hua Chia

**Affiliations:** 1Materials Science Program, Department of Applied Physics, Faculty of Science and Technology, Universiti Kebangsaan Malaysia, Bangi 43600, Selangor, Malaysia; 2Faculty of Engineering, Science and Technology, Nilai University, Nilai 71800, Negeri Sembilan, Malaysia; 3ASASIpintar Program, Pusat GENIUS@Pintar Negara, Universiti Kebangsaan Malaysia, Bangi 43600, Selangor, Malaysia; 4Department of Chemical Engineering, School of Mining, Metallurgy and Chemical Engineering, University of Johannesburg, P.O. Box 17011, Doornfontein 2028, South Africa; 5Sustainable Membrane Technology Research Group (SMTRG), Chemical Engineering Department, Persian Gulf University, Bushehr P.O. Box 75169-13817, Iran; 6Zhejiang Rongsheng Environmental Protection Paper Co. LTD, NO.588 East Zhennan Road, Pinghu Economic Development Zone, Zhejiang 314213, China

**Keywords:** biogenic synthesis, copper-based nanomaterials, extraction method, plant extract

## Abstract

Plants have been used for multiple purposes over thousands of years in various applications such as traditional Chinese medicine and Ayurveda. More recently, the special properties of phytochemicals within plant extracts have spurred researchers to pursue interdisciplinary studies uniting nanotechnology and biotechnology. Plant-mediated green synthesis of nanomaterials utilises the phytochemicals in plant extracts to produce nanomaterials. Previous publications have demonstrated that diverse types of nanomaterials can be produced from extracts of numerous plant components. This review aims to cover in detail the use of plant extracts to produce copper (Cu)-based nanomaterials, along with their robust applications. The working principles of plant-mediated Cu-based nanomaterials in biomedical and environmental applications are also addressed. In addition, it discusses potential biotechnological solutions and new applications and research directions concerning plant-mediated Cu-based nanomaterials that are yet to be discovered so as to realise the full potential of the plant-mediated green synthesis of nanomaterials in industrial-scale production and wider applications. This review provides readers with comprehensive information, guidance, and future research directions concerning: (1) plant extraction, (2) plant-mediated synthesis of Cu-based nanomaterials, (3) the applications of plant-mediated Cu-based nanomaterials in biomedical and environmental remediation, and (4) future research directions in this area.

## 1. Introduction

Apart from food, plants have traditionally been used intensively in textile, cosmetics, and medicine. Beyond traditional Chinese medicine and Ayurveda, which have developed over thousands of years of interest in the prevention and treatment of diseases, the biomedical applications of plants have broadened even further due to the advancement of technology and of time [1,2,3,4]. These biomedical applications are mainly due to the phytochemicals within plants [5,6,7], which are among the most fascinating aspects of plants due to their having activities such as antimicrobial, antitumour, antiaging, and others [8]. Awareness of such properties has driven researchers to discover still more applications of phytochemicals. In 1959, Richard Feynman illustrated the controlling of single atoms and molecules under the topic of “There’s plenty of room at the bottom”, which first shed light on the novel nanotechnology research field. More recently, an innovative interdisciplinary study pioneered nanomaterial synthesis from the phytochemicals within plant extracts, a process which is more eco-friendly than conventional methods and avoids the usage of hazardous chemicals [9,10].

Plant-mediated nanomaterials synthesis is a branch of green synthesis in which the phytochemical compounds in plant extracts are utilised as stabilizing and reducing agents [9,10,11,12,13]. In addition to the pros and cons inherent in the synthesis method, the choice of method and parameters in the nanomaterial production process also affect the geometry of the obtained nanomaterials [14]. Typically, plant-mediated synthesis uses a bottom-up approach for material synthesis from plant extracts with the assistance of different biotechnological methods [11,13,15,16]. Conventional synthesis techniques have their disadvantages, such as use of hazardous chemicals, biological risks, and high energy consumption [11,13,17,18,19,20]. Relative to conventional approaches like physical and chemical synthesis routes, plant synthesis is considered more eco-friendly and less toxic [13].

Many researchers have successfully synthesised, via plant-mediated synthesis, various types of nanomaterials that were previously produced by conventional synthesis approaches, such as alloys, pure metals, metal oxides, and core shells [21,22,23,24,25,26,27,28]. The produced nanomaterials have been used for numerous applications, including as antibacterials, anticancer agents, antifungals, antiparasitics, antioxidants, catalytic reduction agents, catalysts, biosensors, drug delivery vehicles, fuel cells, photocatalysts, and theranostics [24,29,30,31,32,33,34,35]. However, there remain some limitations to the plant-mediated synthesis method that need to be addressed, such as the complexity and diversity of phytochemicals in plant systems, bio-reduction reactions, homogeneity, scaling-up, reproducibility, material accessibility, and product stability [31,36].

Cu is an element that has drawn significant attention from researchers in nanotechnology, specifically in the nanomaterial sector [37]. This is owing to the low cost, good abundance, and conductivity exhibited by Cu as compared to silver (Ag) and gold (Au) [38,39,40,41,42,43]. Accordingly, not only have various Cu nanomaterials (Cu-NMs) been developed, but there is a good body of literature on the plant-mediated synthesis of Cu nanomaterials with various applications [11].

Therefore, this review will focus on the synthesis of Cu-NMs from several perspectives, including their conventional, green, and especially plant-mediated synthesis, and, relatedly, plant extraction methods, parameters of plant-mediated nanomaterials, applications of plant-mediated Cu-NMs, limitations of plant-based synthesis and proposed solutions, and potential new applications and new research directions that are yet to be explored regarding plant-mediated Cu-NMs.

## 2. Synthesis of Nanomaterials: Conventional and Green Approaches

Approaches for the production of nanomaterials can be categorised according to two predominant aspects: top-down and bottom-up [44]. Examples of the subdivisions within each sector will be discussed. Firstly, top-down nanomaterial synthesis methods composed of ball milling and laser ablation, then bottom-up methods including hydrothermal, vapor deposition, microwave, chemical reduction, and green synthesis [45,46].

### 2.1. Disadvantages of Conventional Nanomateiral Synthesis Method

The typical demerits of conventional methods can be observed clearly in the case of ball milling, as it is both energy-intensive and time-consuming to produce nanomaterials by this method, and, hence, ball milling is neither economical nor industry-friendly [45,46,47]. Similarly, the other top-down approach, laser ablation, requires high energy input to produce a sufficiently intense laser for the continuous ablation process [46,48]. On the other hand, among bottom-up approaches, hydrothermal and microwave methods require an expensive autoclave and complex equipment; thus, they are not applicable economically [45,46,48,49], while vapor deposition also necessitates high energy consumption [50]. Meanwhile, chemical reduction utilises many substances that exhibit high toxicity toward living organisms and the environment, such as hydrazine, N, N-dimethylformamide, and sodium borohydride; this results in additional treatment processes also being required [51,52]. Given all of the above drawbacks, many researchers have investigated green synthesis methods in order to discover more biologically friendly alternatives for producing nanomaterials.

### 2.2. Green Synthesis Method of Nanomaterials

Green synthesis methods that utilise natural or biological compounds to produce nanomaterials, such as bacterial-, fungal-, algae-, and plant-based methods, have been found to be non-toxic, non-harmful, and eco-friendly [14,37,47]. The utilization of natural and biologically friendly compounds as reducing or capping agents also offers other advantages such as reducing energy requirements, avoiding usage of toxic/hazardous chemicals, and being simple and cheap [11,13,53]. The synthesis of nanomaterials using bacteria has particular advantages as bacteria is abundant, easily cultured with a short generation time, inexpensive to cultivate, stable, and easy to manipulate at the genetic level [46,54]. On top of that, previous reports have described the adaptability of bacteria to environments with a high concentration of heavy metals via transforming the toxic metal ions to non-toxic metal oxide nanomaterials, which provides another rationale for the utilization of bacteria in nanomaterial production, as precursors could be introduced in higher concentrations [55,56].

In the fungal-mediated synthesis of nanomaterials, fungi show outstanding heavy-metal tolerance, internalization, and bioaccumulation capability, making them good candidates as reducing and stabilizing agents in the synthesis of metal nanomaterials [57]. Moreover, fungi can be reproduced in large quantities, and by the parity of reasoning nanomaterials can be synthesised in quantity [58]. Relative to bacteria, fungi produce higher amounts of proteins and enzymes; thus, they can provide higher productivity of synthesis [46,59].

Algal-mediated synthesis of nanomaterials involves the utilization of carbohydrates, proteins, minerals, lipids, and bioactive compounds within algae as reducing agents to reduce metal precursor ions into nanomaterials [60]. Given their heavy-metal hyperaccumulation capability, algae are excellent candidates for nanomaterials synthesis [61,62]. The algae-mediated production of nanomaterials can occur either via extracellular or intracellular processes and affords good control over production parameters [60]. However, although the various microorganism-mediated synthesis processes offer many benefits, the pathogenic properties of organisms, underlying safety concerns, and deficit of knowledge regarding synthesis mechanisms are drawbacks that yet hinder the use of these processes in industrial nanomaterial production and applications [46,54,63].

Plants are particularly good candidates for nanomaterial synthesis since they have no pathogenic effects as microorganisms do, plus the nanomaterials produced via plant biogenic synthesis are more homogenous in comparison to the products of other methods [46,54]. In addition, unlike other synthesis methods, the mechanism of plant-mediated metal nanoparticle synthesis is limited to the reduction of a precursor salt via agents within the plant extract in the presence of a metal ion precursor. Moreover, stabilizing agents within a plant extract can also attach to the surface of the produced nanoparticles, improving the surface reaction kinetics as well as particle stability and, hence, reducing the deformation and agglomeration of particles [64]. The reducing and stabilizing agents that participate in the formation of nanomaterials consist of phytochemicals such as amino acids, proteins, vitamins, terpenes, flavones, ketones, amides, saponins, phenolics, terpenoids, aldehydes, alkaloids, carboxylic acids, and polysaccharides naturally found within the plant [11,13].

## 3. Plant-Mediated Nanomaterial Synthesis

The most essential element in plant-mediated nanomaterial synthesis is the plant extract. While a number of approaches have been developed for obtaining extracts, the overall technique can be generalised into the few steps illustrated in Figure 1.

Notably, extracts can be obtained from multiple different parts of plants, including leaves, fruits, peelings, flowers, rhizomes, roots, and seeds; see Table 1.

### 3.1. Plant Extraction Method

The first step of plant extraction is the cleaning process, which mainly aims to remove debris or dust with water so as to avoid any form of contamination that might affect the subsequent synthesis process. The second step consists of drying and downsizing. Drying is necessary to avoid the deterioration of phytochemicals that results from enzymatic and microbial activities due to the presence of water moisture [160].

#### 3.1.1. Drying

Typically, drying is performed via air drying, shade drying, oven drying, drying in a dehydrator, vacuum drying, sun drying, or on filter paper; plant materials can also be acquired in the dry form (Table 1 and Figure 1).

Each of the abovementioned drying methods is able to successfully yield plant extracts with phytochemicals. Shade and air drying are considered among the best methods as they allow the greatest preservation of nutrients, such as proximate and ascorbic acid, and do so with lower financial cost as compared to mechanical drying methods such as oven drying, vacuum drying, or using a food dryer [160]. As a case in point, tangerine peel was shade dried at 27 ± 2 °C for the synthesis of iron oxide nanoparticles [116]. However, due to being carried out at a lower temperature, shade and air drying require a longer period of time than other drying methods, which might reduce their applicability in the industrial plant-mediated synthesis of Cu-NMs [160]. For instance, in preparation for Au nanoparticle synthesis, *Nepeta leucophylla* root was shade dried at room temperature (24–32 °C) for 30 days [137]. Sun drying was also used in drying *Chromolaena odorata* for the synthesis of Fe_3_O_4_ nanoparticles from phenolic components of the extract [132]. While sun drying can reduce the cost of drying just as can shade drying, it is not recommended for industrial synthesis due to high labour demand, low efficiency, hygiene issues, and more precautions being required to avoid contamination of samples [160].

The temperature of the drying process also plays a major role in preserving the phytochemicals within a plant. Specifically, drying temperatures in the range of 40–60 °C are reported to support the minimal loss of phytochemicals in plant components [160]. In prior studies, neem leaves (*Azadirachta indica*) were oven dried for 15 min at 50 °C [65], and *Garcinia mangostana* peelings for 10 min at 40 °C [109]. Although the range of 40–60 °C is recommended, the final decision on which temperature is most suitable for the drying process should be based on the characteristics of the plant material being dried. For example, a study oven dried *Arachis hypogaea* at 70 °C for 30 min due to its anthocyanin content, which is highly preserved under those drying parameters [105]. Nonetheless, drying at room/ambient temperature remains the most used method owing to the low cost requirement being beneficial to industrialization. For instance, Irum et al. [69] shade dried *C. jwarancusa* at room temperature while Elgorban et al. [123] dried calendula flowers at room temperature to acquire phytochemicals. This is despite the time requirement being much higher; for instance, when drying at room temperature, Yulizar et al. [159] took a week to dry *Theobroma cacao* seeds and Rautela et al. [158] 3–4 days for *Tectona grandis* seeds in preparation for nanoparticle synthesis. On the other hand, Pan et al. [105] only need 30 min to dry *Arachis hypogaea* with an oven at 70 °C in plant-mediated iron nanoparticle synthesis, while Doan Thi et al. [111] took 12 h to dry orange peels for ZnO nanoparticle production.

In addition to the abovementioned drying methods, some plant components simply are not subjected to any drying process, mainly in the interest of cost saving and because certain components have high water contents that will increase the cost if a drying process is applied. Such plant components can include fruits, flowers, seeds, roots, and rhizomes. In one example, Jahan et al. [90] squeezed the juice from *Citrus sinensis* fruits to acquire reducing sugars, amino acids, proteins, and metabolites such as flavanones and terpenoids for the synthesis of Cu nanoparticles. The same squeezing method was also applied to *Zingiber officinale* root by Velmurugan et al. [144] to acquire alkaloids and flavonoids for the synthesis of Au and Au nanoparticle. Crushing is another technique for acquiring plant extracts; for example, Kumari et al. [153] crushed pomegranate seeds to obtain flavonoids and terpenoids for the synthesis of Au-Ag bimetallic nanoparticles. Moreover, some methods forgo any drying treatment, such as when Patra et al. [127] directly extracted *Muntingia calabura* flowers to acquire phytochemicals for nanoparticle synthesis and Al-Radadi [135] used licorice root without drying to obtain glycosides, organic acids, phenolic compounds, and flavonoids for the synthesis of Au nanoparticles. Ultimately, the characteristics of the plant component being used and the potential cost are important factors informing the best drying method and parameters by which to obtain the most phytochemicals from plant components for Cu-NM synthesis for either research or industrial purposes.

#### 3.1.2. Downsizing

Regarding the downsizing step, its primary purpose is to reduce the size of the plant components and increase their surface area, leading to better diffusivity and mass transfer in order to extract the greatest yields of phytochemicals such as polyphenolic compounds, phenolic acids, and tannins [161]. There are various routes for achieving this objective, presented in Table 1 and Figure 1. Interestingly, miniscule deviations of plant component size can cause significant alterations in overall phytochemical yield [161]. Therefore, it is necessary to consider carefully the most suitable methods and cost requirements so as to acquire the smallest plant components with the highest phytochemical yields for Cu-NM synthesis. Just as with the drying process, there are some plant components that do not undergo any downsizing, such as those with high water content; for example, *Crataegus pentagyna* fruits were extracted by Ebrahimzadeh et al. [93]without any downsizing.

#### 3.1.3. Plant Extraction Methods

Plant extraction methods are mainly based on boiling and heating (Table 1). Mani et al. [66] conducted an extraction from dried, ground, and pulverised *Basella alba* leaves by mixing them with DI-H_2_O and boiling them in a water bath at 60 °C for 20 min. Nnadozie and Ajibade [132] similarly heated crushed *Chromolaena odorata* at 85 °C for 2 h in DI-H_2_O, and Abisharani et al. [148] heated *Cucurbita pepo* seeds with DS-H_2_O at 90 °C for 2 h.

Interestingly, some alternative methods have been introduced and successfully used to extract phytochemical products from plants (Table 1). For example, Siddiqui et al. [113] boiled powdered *Punica granatum* peels in sterile DI-H_2_O at 55 °C for 30 min on a Soxhlet apparatus, and Singh and Dhaliwal similarly performed Soxhlet extraction on powdered *Nepeta leucophylla* roots with methanol held at boiling for 8 h [137]. Sonication has also been used in plant extractions; for instance, one study removed the coats of *Caesalpina bonducella* seeds and then sonicated the ground kernels for 30 min [146]. Likewise, reflux extraction has been used with various plant components. Beheshtkhoo et al. [70] extracted *Daphne mezereum* leaves by refluxing the dried leaves with a 5% (*w*/*v*) mixture in DI-H_2_O for 15 min. Microwave irradiation has also been used in extraction, such as in a study that irradiated cut *Jasminum sambac* leaves in DS-H_2_O for 200 s to extract phytochemicals for the synthesis of Au, Ag, and Au−Ag alloy nanoparticles [74]. Maceration has also been used by many researchers, mainly due to its low cost and eco-friendliness; for instance, ground *Solanum mammosum* fruits were macerated with DI-H_2_O at room temperature and constant agitation for 1 h [100] and *Crataegus pentagyna* fruits with methanol at room temperature for the synthesis of Fe_3_O_4_-SiO_2_-Cu_2_O–Ag nanocomposites [93]. In addition to the above, autoclaving was carried out on dried and ground roots of *Scutellaria baicalensis* with DS-H_2_O for 30 min at 100 °C in preparation for the synthesis of ZnO nanoparticles [142].

Aside from single extraction methods, combinations of methods have also been applied to acquire extracts from various plants. For example, Nava et al. [117] macerated the peels of *Citrus aurantifolia*, *Citrus paradisi*, *Citrus sinensis* and *Lycopersicon esculentum* for 3 h with stirring, then heated the mixture at 60 °C for 60 min. For plant components with high water content, a squeezing method may be introduced. For example, in the preparation of *Zingiber officinale* root extract by Velmurugan et al. [144], the downsized roots were squeezed via muslin cloth.

Every extraction method has its pros and cons, summarised in Table 2 and Figure 1 [4,161,162]. The most suitable method for any given use case depends on the types of plant components as well as the requirements and restriction posed by the actual environment, such as a need to reduce financial and labour costs for industrial purposes as well as a requirement for eco-friendliness.

##### Solvents in Plant Extraction

In addition to the extraction method used, solvent, energy consumption, time required, and other parameters are also critical to the extraction of phytochemicals [161,163,164]. Extraction solvents can be divided into two types. i.e., water (distilled, double distilled, Milli-Q, ultra-pure, and deionised) and alcoholic solvents (ethanol and methanol), as presented in Table 1. Water (DS-H_2_O) was used to extract dried and ground *Quercus coccifera* leaves with boiling for 30 min at 90 °C [80]. Conversely, Boruah et al. [77] produced *Moringa oleifera* leaf extract by Soxhlet extraction with methanol as the solvent, incubating the dried and powdered leaves at 35–45 °C for 10 h. Some phytochemicals, such as polyphenolic compounds, anthocyanins, and polyphenols, can be obtained at higher yields when an alcoholic solvent is involved. Conversely, Do et al. [165] found that the phytochemical extraction yield from *Limnophila aromatica* improves as the solvent polarity increases; in particular, methanol could extract more phytochemicals than ethanol. There are also some cases that benefit from extraction solvents combining both water and an alcoholic solvent. For example, *Piper longum* fruits were dried, powdered and extracted with 30% methanolic solution at 70 °C for 30 min [97], and Zarei et al. [89]used ethanol and water at a 1:1 ratio with boiling for 30 min. Remarkably, such combinations of alcoholic solvents with water can achieve the highest yields due to allowing for greater solubility of plant components [161,165,166]. Therefore, it could be concluded that for plant phytochemical extraction, an alcoholic aqueous solvent is generally the most suitable. Nonetheless, the specific characteristics of the plant and phytochemicals should be considered before applying a particular type of solvent. As a case in point, Maurya et al. [145] produced *Bixa orellana* seed extract using ethanol mainly due to the primary phytochemical *cis*-bixin being water insoluble.

##### Temperature in Plant Extraction

The temperature applied is also a crucial factor in the plant extraction process as it can greatly impact the yield and quality of phytochemicals and, thus, affect the nanoparticle synthesised. As listed in Table 1, the temperature for extraction may range from room temperature to 100 °C. As an example of room-temperature extraction, Pilaquinga et al. [100] subjected pre-washed, oven-dried, and ground *Solanum mammosum* fruits to maceration with DI-H_2_O at room temperature with constant agitation for an hour, while, as an example of the highest temperature, Hu et al. [141] extracted *Rhodiola rosea* rhizome powder by heating with DI-H_2_O at 100 °C for 30 min. The temperature applied has a directly proportional relationship with solubility and diffusion. Nevertheless, when the temperature surpasses a particular threshold, it might lead to several problems such as solvent loss, introduction of impurities in the produced extract, and decomposition of thermolabile phytochemicals. For instance, when synthesizing Ag and Au nanoparticles from *Impinella anisum* seeds extracted at temperatures ranging from 25 to 60 °C, high surface plasmon resonance (SPR) peak intensities accompanied the raising of temperature due to the increased diffusion rate of the solvent, which destroyed the plant cell structure. However, when temperatures in the range of 60 to 85 °C were used, reduction in SPR was observed due to the decomposition of some thermolabile phytochemicals [167].

##### Extraction Time in Plant Extraction

Extraction time is another synergic factor that can greatly affect the phytochemicals extracted. Durations reported in the literature range from 200 s to a week; in addition, it can also be observed that the higher the temperature applied, the lower the extraction duration, and vice versa (Table 1). At the short end, Yallappa et al. [74] conducted an extraction of *Jasminum sambac* leaves in DS-H_2_O assisted by microwave irradiation for 200 s. Meanwhile, for the longest duration, Yulizar et al. [159] macerated *Theobroma cacao* seed bark powder in methanol with stirring for a week. Extending the extraction duration can improve extraction efficiency as the mass transfer coefficient between plant components and solvent increases; accordingly, longer extractions can boost the quantities of extracted phytochemicals and so enhance the formation of subsequently synthesised nanoparticles. However, such phenomena are restricted to within a certain time range, as when equilibrium has been reached inside and outside of the plant components, the extraction efficiency will not be further improved and could even worsen if the extraction period is excessively prolonged [4,167]. For example, extraction of *Impinella anisum* seeds for 60 min results in the greatest band intensity for subsequently produced nanoparticles, and band intensity then declines as the extraction duration increases due to the oxidation and thermal decomposition of phytochemicals [167]. Therefore, attentive consideration should be made regarding the duration of, and temperature during, phytochemical extraction.

##### Filtration and Preservation

After extraction, the next step is filtration, in which solid components are removed from the plant extract. There are many filtration techniques in use, as illustrated in Figure 1.

Following filtration, the obtained extracts are preserved for nanomaterial-synthesis research. Preservation is mainly achieved via refrigeration, directly using the extract for nanoparticle synthesis, or storing the extract in a container/environment with or without special conditions such as airtightness and light exclusion so as to avoid any manner of the oxidation or photodegradation of the phytochemicals. The temperature of refrigeration is mainly 4 °C as it was found that this temperature can best preserve the quality of *Ananas comosus* juice; moreover, increasing storage duration and temperature can greatly reduce the phytochemicals within the obtained plant extract [168]. Therefore, in the green synthesis of Cu-NMs, the freshness of the plant extract is very significant. Once a plant extract is produced, it should be utilised for nanoparticle synthesis as soon as possible and, in the interim, stored at low temperature.

Finally, the obtained plant extract is prepared for the synthesis of nanomaterials; for example, Nasrollahzadeh et al. [85,169] produced *Thymus vulgaris* leaf extract and used it to synthesise CuO and Cu nanoparticles, as shown in Figure 2.

It is worth knowing that, although the production of other type of nanomaterials via green synthesis methods are referenced in this review, the plant extraction methods mentioned above are compatible in Cu-NMs synthesis.

Although the above paragraphs generalised the parameters and methods for plant extraction, there is no one best universal extraction method and parameter set for extracting all phytochemicals from all plant components. The final selections should depend on the type of plant, the plant component, and any industrial requirements.

Next, this review covers the synthesis of Cu-NMs using plant extracts. There are several factors that need to be taken into account to ensure the successful production of nanomaterials, including reaction time, temperature, pH, and the extract/precursor used; these will all influence the size and geometry of the nanomaterial produced. Table 3 summarises previously reported works on the synthesis of Cu nanomaterials using plant extracts.

**Figure 3 nanomaterials-12-03312-f003:**
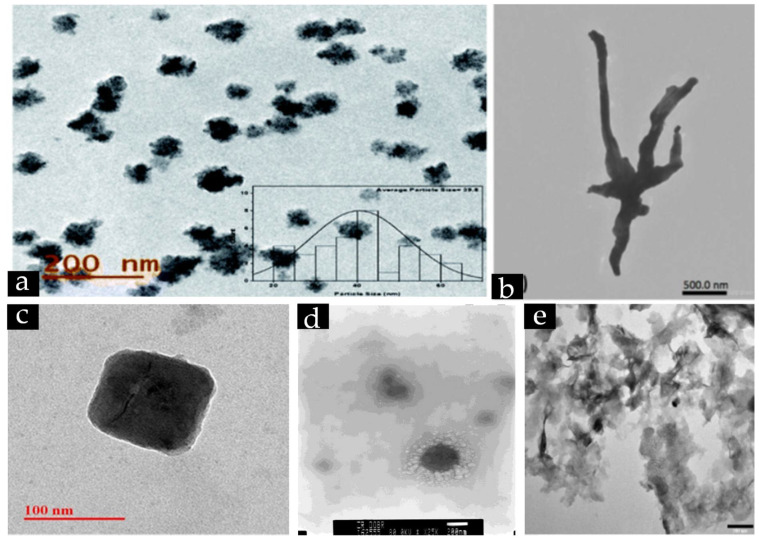
Representative TEM images of (**a**) spherical CuO nanoparticles synthesised using *Annona squamosa* seed extract. Reproduced from [204]. 2021with permission from the Royal Society of Chemistry, (**b**) tentacle-like bimetallic Ag-Cu nanoparticles synthesised using *Carica papaya* extract. Adapted with permission from Ref. [172]. 2017, Elsevier, (**c**) cubical Cu nanoparticles synthesised from *Azadirachta indica* leaf extract Adapted with permission from Ref. [65]. 2018, Elsevier; (**d**) SEM image of spherical Cu-Pt core shell nanoparticles synthesised using *Agrimoniae herba* extract. Adapted with permission from Ref. [170]. 2018, Elsevier; and (**e**) TEM image of Cu-Co-Ni trimetallic nanoalloy nanoflakes synthesised using *Origanum vulgare* leaf extract. Adapted with permission from Ref. [176]. 2020, MDPI.

### 3.2. Cu-NMs Synthesis Method

#### 3.2.1. Production of High Tunable Cu-NMs

All extracts from plant components are composed of various types of phytochemicals such as flavonoids, phenolics, alizarin, quercetin, terpenoids, terpenes, alkaloids, carotenoids, and others. These phytochemicals can potentially be used for the synthesis of various types of Cu-based nanomaterials such as pure Cu nanoparticles, CuO nanoparticles, Cu-based nanocomposites, core-shell nanoparticles, and nanoalloys. For example, Ituen et al. [186] used *Citrus reticulata* peel extract to produce spherical Cu nanoparticles with sizes of 54 and 72 nm. Similarly, *Azadirachta indica* flower extract has been used to produce pure spherical 5 nm Cu nanoparticles [122], and *Thymus vulgaris* leaf extract to produce CuO nanoparticles with spherical morphology and a particle size less than 30 nm [85]. In addition, Dobrucka and Dlugaszewska synthesised 30 nm spherical bimetallic Pt-Cu core-shell particles with Cu as the core and Pt as the shell from the ethanolic extract of *Agrimoniae herba* leaves [170]. As an example of root-extract-mediated nanoparticle synthesis, Pallela et al. [192] successfully produced rod-shaped CuO nanoparticles with diameters of 50–100 nm and lengths of 400–500 nm from *Asparagus racemosus* root extract. Regarding nanoparticle synthesis from fruits, Ebrahimzadeh et al. [93] produced spherical Fe_3_O_4_/SiO_2_/Cu_2_O–Ag nanocomposites with diameters of 55 and 75 nm from *Crataegus pentagyna* fruit extract. Meanwhile, Sajadi et al. [201] used *Silybum marianum* seed extract to produce agglomerated Cu/Fe_3_O_4_ nanoparticles with sizes of 8.5–60 nm and magnetic properties.

As indicated in Table 3, the sizes of nanoparticles produced using plant-mediated synthesis ranges from 2 to 150 nm. On the other hand, the morphology of the produced nanoparticles is predominantly spherical [172,205,206] with some having other shapes such as hexagonal, cubical [65,71], ellipsoidal [175], tentacle-like [172], or nanoflake [171,176] (Table 3 and Figure 3). Several characterization methods are used in determining the size and morphology of nanoparticles, with the preeminent being scanning electron microscopy (SEM), transmission electron microscopy (TEM), and dynamic light scattering (DLS). It should be noted that the findings elucidated by each characterization method might have some discrepancies. For instance, Rosbero and Camacho observed synthesised Ag/Cu nanoparticles to have a size of 420.70 nm according to DLS, but a size range of 90–150 nm by TEM [172]. This discrepancy is attributable to the presence of solvent molecules on the nanoparticle surface and DLS only determining the hydrodynamic size of the particles rather than the core diameter. That is to say, when there is a hydration layer surrounding the nanoparticles, only the solvated particle size is indicated by a particle’s diffusional characteristics [172,186]. Hence, to obtain the most accurate results and perform effective quality control of nanomaterials, multiple characterization methods should be employed.

Overall, it can be observed that Cu-NMs produced via plant-mediated synthesis feature size and morphology tunability comparable to those obtained with chemical synthesis methods. Specifically, tunability can be achieved via altering parameters such as the precursor concentration, plant extract, reaction time, and the temperature applied during nanoparticle synthesis.

#### 3.2.2. Precursor

As listed in Table 3, most studies to date have utilised CuSO_4_ [188,202,205], Cu(NO_3_)_2_ [120,186,194], and CuCl_2_ [79,199] as the Cu precursor, while some used copper acetate (Cu(OAc)_2_) [71,181]. Interestingly, for some multi-metallic nanoparticles (nanoalloys, core-shell particles, and nanocomposites), multiple precursors have been utilised and the type of nanoparticle formed depends on the methodology. For example, Cu-Co-Ni trimetallic nanoalloy was synthesised using *Origanum vulgare* leaf extract and the precursors of Cu(NO_3_)_2_·3H_2_O, Ni(NO_3_)_2_·6H_2_O, and Co(NO_3_)_2_·6H_2_O [176], while bimetallic Pt-Cu core-shell structures with Cu as core and Pt as shell were synthesised from *Agrimoniae herba* leaf ethanolic extract with K_2_PtCl_6_ and CuSO_4_ as the precursors [170]. Basically, multiple metallic precursors are mixed with a plant extract and stirring and heat applied to yield multi-metallic nanoparticles. Generally, a combination of two metals will lead to the synthesis of alloy or core-shell nanoparticles. Which form of Cu-NM is produced can be determined based on SPR from UV–visible analysis: if a single SPR is found, an alloy was formed, while if two independent and continuous peaks are evident, a core-shell-type structure resulted [172]. For instance, Ag/Cu nanoparticles produced via bio-reduction with *Carica papaya* extract exhibited a single peak at 776 nm as the maximum absorption, suggesting an alloyed structure [172]. Cu-based nanocomposites are also able to be synthesised via plant-mediated methods; Ebrahimzadeh et al. [93] produced spherical Fe_3_O_4_/SiO_2_/Cu_2_O–Ag nanocomposites of 55 and 75 nm in size using *Crataegus pentagyna* fruit extract with FeCl_3_·6H_2_O, FeCl_2_·4H_2_O, Ag(NO_3_), Cu(NO_3_)_2_·3H_2_O, and tetra ethyl orthosilicate.

In addition to precursor choice, the concentration of precursor applied in the reaction also plays an important role in determining the Cu-NMs synthesised. Lee et al. [75] used *Magnolia kobus* leaf extract to produce spherical Cu nanoparticles, mixing it with CuSO_4_·5H_2_O at 0.5, 1, and 2 mmol/L and reducing the Cu ions to atoms. Given constant temperature and plant extract concentration, the time required to achieve a conversion rate of more than 90% was 1600, 1400, and 200 min for the concentrations of 0.5, 1, and 2 mmol/L, respectively. Therefore, it could be concluded that a higher precursor concentration can accelerate nanoparticle formation. In addition, a study that performed precursor optimization for the green synthesis of Cu nanoparticles from *Senna didymobotrya* root extract utilised CuSO_4_·5H_2_O at concentrations of 0.0125, 0.03125, and 0.05 M [196]. This study revealed that the higher the precursor concentration, the higher the nanoparticle size. The authors noted this could be due to a low concentration of Cu ions reducing the chance of Cu-Cu interactions and, hence, reducing agglomeration [196]. Thus, the formation rate and size of synthesised nanoparticles can be controlled via altering the precursor concentration. However, the balance between conversion rate (nanoparticle formation) and nanoparticle size should be taken into account when carrying out green-synthesis research to ensure the desired nanoparticle is produced while also achieving a highly productive and efficient synthesis process.

#### 3.2.3. Plant Extract

The plant extract utilised is also another major factor that should be considered in plant-mediated Cu-NM synthesis. A variety of plant extracts have demonstrated great impact on the synthesis of Cu-NMs (Table 3). In a study examining the effect of extract concentration on nanoparticle synthesis rate and characteristics, *Magnolia kobus* leaf extract at a range of concentrations (5–20%) was used to produce spherical Cu nanoparticles [75]. The highest synthesis rate was obtained with an extract concentration of 20%, while high average particle sizes were obtained for both the lowest (5%) and highest (20%) extract concentrations, with diameters of 91 and 82 nm, respectively. Meanwhile, an extract concentration of 15% produced nanoparticles with diameter 37 nm, which was the smallest among all the results [75]. The reason for the production of large nanoparticles from high extract concentrations is due to the excessive abundance of capping materials promoting the aggregation of Cu particles owing to the interaction between nanoparticles that are surrounded with proteins and metabolites (reducing sugar, terpenoid, and other metabolites) [75]. Therefore, it can be concluded that if a high yield of a small Cu nanoparticle is required, the leaf extract concentration should be optimised before conducting the plant-mediated synthesis process at scale, whereas if the most rapid production is required, a high concentration of plant extract should be applied.

#### 3.2.4. Temperature

Aside from material inputs, the temperature applied during plant-mediated Cu-NM synthesis is also an essential parameter to be investigated. According to Table 3, the temperatures utilised in existing reports range between 25 and 100 °C. For example, temperatures of 25 ± 2, 30, 40, and 50 °C have been used for Cu nanoparticle production from *Citrus reticulata* peel extract [186], and 100 °C for CuO nanoparticle synthesis from *Rosa canina* fruit extract [181]. Notably, the use of different temperatures can greatly impact the Cu-NM synthesis process. For example, in the abovementioned study using *Citrus reticulata* peel extract combined with CuSO_4_.5H_2_O [186], successful bio-reduction and nanoparticle production was indicated by a colour change to brown with absorption at 442 nm (Figure 4). At reaction temperatures of 25, 30, 40, and 50 °C with a constant pH, achieving this endpoint required 72 h, 60 h, 10 h, and 105 min, respectively. Therefore, plant-mediated synthesis of Cu-NMs is a temperature-dependent process with a positive proportional relationship: the higher the temperature, the higher the rate of conversion from Cu ion to Cu metal.

Interestingly, nanoparticle synthesis rate and size behave differently under a given reaction temperature increment, as the conversion rate increases whereas nanoparticle size decreases with increasing reaction temperatures. For instance, as mentioned above, Lee et al. [75] synthesised Cu nanoparticles using *Magnolia kobus* leaf extract and observed a size reduction from 110 nm at low temperature (25 °C, conversion rate 70%) to 37 nm at high temperature (95 °C, conversion rate ~80–100%). The rationale behind such phenomena is that the increasing temperature improves the reaction rate. When the reaction rate is increased, Cu ions in the reaction solution are only able to be consumed for the formation of nuclei; the secondary reduction process on the nuclei is avoided. Thus, larger nanoparticles cannot be produced at higher temperatures [75,196]. Consequently, it can be concluded that nanomaterials synthesis is temperature-dependent, but the size of nanomaterials has a negative proportional relationship with temperature such that producing nanoparticles with a larger size necessitates utilising a lower temperature.

There have been studies conducted on further calcination of metal oxides at temperatures ranging from 400 to 500 °C after the synthesis process (Table 3) [27,179,190]. One of the purposes of calcination is to produce stable metal oxides or metal oxide nanocomposites through oxidation [177]. For example, Suresh et al. [177] used *Pisonia grandis* leaf extract to synthesise Zn-Mg-Cu oxide nanocomposites, then calcinated them at 450 °C to obtain mixed metal oxide nanocomposites. In addition to producing stable oxides, increasing calcination temperature can boost the size of Cu-NMs and produce black precipitates of agglomerated cubical nanomaterials, where uncalcinated nanomaterials have elongated morphology [205].

#### 3.2.5. pH

Solution pH is also a very significant factor in plant-mediated nanomaterial synthesis as it can affect the synthesis rate and products. Mechanistically, the importance of pH is due to the reducing and stabilizing agents being greatly dependent on the phytochemicals within the plant extract, which might be readily affected by pH. Generally, the best pH values for plant-mediated nanomaterial synthesis are in the range of pH 7–9, and varying the pH will alter the nanoparticle synthesised [14]. Nagar and Devra utilised *Azadirachta indica* leaves for Cu nanoparticle synthesis at various pH values [65]. They found that nanoparticle synthesis is more effective at higher pH and abolished in an extreme acidic environment, such as pH 4.7. A solution with pH 6 produced small-sized nanoparticles of 56 nm, while in an alkaline environment of pH 9.3, the nanoparticles produced were of size 73 nm. In an acidic environment, the phytochemicals in the plant extract might be inactivated [65]. In addition, lower pH can cause nanoparticles to experience high electrostatic repulsion which reduces the chances of agglomeration and, thus, yields nanomaterials of smaller size [196]. Conversely, in a more alkaline condition such as pH 10, the low electrostatic forces of the nanoparticles allow further particle growth and agglomeration, which produces larger nanomaterials [65,196]. It is worth noting that during the plant-mediated synthesis process, the pH of the medium will drop as the Cu^2+^ ions cause oxidation of the plant extract, leading to the release of H^+^ ions and, hence, acidification; this is also another important aspect to be considered by researchers carrying out plant-mediated nanomaterial synthesis [65]. Therefore, it is necessary to achieve a balance in producing nanoparticles with a desired size while maintaining high nanomaterial productivity.

#### 3.2.6. Reaction Time

In terms of duration, it can be seen that the range of reaction times in the literature is relatively large, ranging from as low as 5 min to as long as 72 h. The duration is not as impactful compared to the other parameters mentioned above [205]. Although a long synthesis duration allows improvement in the nanomaterial nucleation rate, the reaction rate will not continue to increase after the optimum time has been reached. In some cases, prolonging the incubation might even cause nanomaterial aggregation [206]. In fact, other parameters such as temperature, precursor, and type of plant extract can greatly impact the time needed to achieve complete conversion from metallic ions (Cu and other ions depending on the type of nanomaterial being produced) to metallic atoms and, finally, nanomaterials, as was mentioned previously in relation to other parameters. However, compared with other materials such as Au, Ag, and Pt, Cu forms nanoparticles relatively slowly as the initiation of Cu nucleus formation is much more difficult [75]. Hence, the green synthesis of Cu nanoparticles necessitates longer reaction times in order to achieve 100% conversion.

#### 3.2.7. Indication of Cu-NMs Production

During the production of Cu-NMs, a successful reaction is indicated by the colour alteration of the reaction solution. For example, *Thymus vulgaris* leaf extract mixed with CuCl_2_.2H_2_O with constant stirring at 60 °C undergoes a change in colour from yellow to dark brown, as shown in Figure 2. In addition, calcination will also change the colour of nanomaterials produced. *Carica papaya* peel extract combined with Cu(NO_3_)_2_.3H_2_O and heated at 70–80 °C changes colour from greenish-blue to green and finally produces a dark green paste. Upon calcination, a fine black-coloured powder was obtained and harvested as CuO nanoparticles [26].

## 4. Applications of Cu-NMs from Plant-Mediated Synthesis

After the synthesis reaction is completed, the obtained nanoparticles are washed, dried, and employed in applications. Cu-NMs synthesised using plant extracts have been utilised in two major areas, namely, biomedical and environmental remediation [31].

### 4.1. Biomedical

Plant-mediated Cu-NMs have demonstrated antimicrobial, antioxidant, and anticancer activities, and have potential as nano-sensors and in various medical applications. In this section, details and mechanisms pertaining to this area will be discussed.

#### 4.1.1. Antimicrobial

Firstly, antibacterial activity has been observed for plant-mediated Cu-NMs [205] and can be attributed to several putative pathways. Bhavyasree and Xavier suggested that Cu-NMs, including both Cu and CuO nanoparticles produced via plant-mediated synthesis, can carry out antibacterial activity through a chemisorption-based mechanism [206]. This mechanism involves microbial adsorption to the nanoparticle surface, which has been bio-functionalised by phytochemicals during the plant-mediated synthesis process. The adsorption is mainly due to chemisorption via non-electrostatic forces (Van der Waals force and hydrogen bonding), which causes the destruction of the microbial cell wall and subsequent cell membrane damage, DNA breakage, and eventually cell death, as illustrated in Figure 5.

Another antibacterial mechanism is mediated by reactive oxygen species (ROS) and the release of Cu^2+^ ions [207]. First, the CuO nanoparticles are much smaller (being of a nanometre scale) than the micrometre-scale pores of bacterial cells, which allows them to easily penetrate the cells. In addition, Cu^2+^ ions are attracted toward bacterial cells due to the abundance of carboxyl and amine groups on the cell surface; this is another factor in antibacterial ability. However, the antibacterial interactions are different for Gram-positive and Gram-negative bacteria, as described in Figure 6a.

After bypassing the cell wall, Cu^2+^ ions relocate intracellularly to the cytosol due to the internalization of CuO nanoparticles and Cu^2+^ ions, where they cause ROS to accumulate [205]. Consequently, DNA and mitochondria damage occur. Cu^2+^ ions within a bacterium may also stimulate cellular responses that lead to bactericidal activity. For example, radicals produced by CuO nanoparticles, such as superoxide and hydroxyl radicals, can have synergic effects in causing bacterial membrane destruction, DNA damage, attachment to ribosomes, oxidative injury, and protein and proton efflux pump damage; they can also prevent biofilm production [205,207].

Cu-NMs exert antibacterial activity through mechanisms similar to those of Cu and CuO nanoparticles. Generally, the antibacterial activity of nanomaterials is mainly owed to the induction of oxidative stress, such as through the production of free radicals and ROS. Notably, nano-sized particles will feature a smaller surface-to-volume ratio, harbour more surface defects due to oxygen vacancies, and feature greater electrostatic attraction and release of Cu^2+^ ions and generate more oxidative stress within the bacterial cells [178]. In addition, bimetallic nanoparticles can demonstrate a synergic effect with improved antibacterial ability. For example, bimetallic Ag and Cu nanoparticles produced by *Vitex negundo*-mediated synthesis demonstrate antibacterial activity when applied in a cellulose matrix via the disc method against both Gram-positive (*Escherichia coli*, *Pseudomonas*, *Klebsiella*) and Gram-negative (*Staphylococcus*, *Bacillus*) species [178]. Particles having an equal ratio of Ag and Cu (2.5 mM each) exhibited the greatest antibacterial ability, with a 9 mm zone of inhibition for all the tested species.

Secondly, Cu-NMs have also been demonstrated to possess antifungal activity. A number of fungi can cause infections in humans with severe symptoms, such as *Candida albicans* which can cause mucosal infections (oropharyngeal or vulvovaginal candidiasis), or *Trichophyton mentagrophytes* which can cause dermatophytosis [208,209]. Antifungal activity is more challenging to realise than antibacterial as a fungus cell has several layers of lipids within its cell wall which impede the penetration and internalization of Cu nanomaterials [205]. Although fewer publications exist regarding the antifungal testing of Cu nanomaterials synthesised by green methods, their hypothesised antifungal mechanism is based on altering the structure and function of fungal cell components [210]. That is, the nanoparticles first distort the cell wall and become internalised by the fungus (Figure 6b). After internalization, the same process of ROS generation and subsequent process disruption ensues as in bacteria, impacting DNA, mitochondria, replication, protein synthesis and other essential elements, eventually leading to cell death [205,210].

In one report of Cu-NM antifungal activity, Mali et al. [210] tested the efficacy of Cu nanoparticles derived from *Celastrus paniculatus* leaf extract against *Fusarium oxysporum*. Concentrations of 0.12%, 0.18% and 0.24% (*w*/*v*) Cu nanoparticles were found able to inhibit mycelial growth by 76.29 ± 1.52%, 73.70 ± 1.52%, and 59.25 ± 0.57%, respectively, calculated via the following formula:(1)$$\left(\%\;inhibition\;rate\right)=\frac{\left(Mc-Mt\right)}{Mc}\times 100$$
where *Mc* represents mycelial growth in the control (with water) while *Mt* is mycelial growth under the Cu nanoparticle treatment. The inhibition rate was found to be dosage-dependent: the higher the Cu nanoparticle dosage, the higher the degree of inhibition.

Dobrucka and Dlugaszewska similarly studied the antibacterial and antifungal activities of Cu-Pt core-shell nanoparticles synthesised using *Agrimoniae herba* extract [170]. The nanoparticles were applied via the well-diffusion method to three species of bacteria, including *Staphylococcus aureus*, *Escherichia coli*, and *Pseudomonas aeruginosa*, and three of fungi: *Candida albicans*, *Trichophyton mentagrophytes*, and *Aspergillus fumigatus*; the authors then determined the minimal inhibitory concentration (MIC), minimal bactericidal concentration (MBC), and minimal fungicidal concentration (MFC) [170]. The Cu-Pt nanoparticles exhibited good inhibitory function on all tested bacteria and *Trichophyton Mentagrophytes*. The overall best antibacterial and antifungal performances were obtained on *Staphylococcus aureus* (MIC of 16.7 and MBC of 33.3) and *Trichophyton mentagrophytes* (MIC and MFC of 26.7).

From the above reports, it can be concluded that plant-mediated Cu-NMs are suitable as antibacterial (for both Gram-positive and -negative) and antifungal agents. Such characteristics are useful in further broadening the application of Cu-NMs in the pharmaceutical and medical sectors.

In addition to direct antimicrobial effects, many plant-mediated Cu-NMs have also demonstrated antioxidant properties which also contribute to antibacterial and antifungal activities as a synergic factor [205]. Multiple mechanisms contribute to antioxidant ability, which are: (1) binding of transition metal ion catalysts, (2) reductive capacity, (3) radical scavenging activity, (4) decomposition of peroxides, (5) prevention of continued hydrogen abstraction, and (6) prevention of chain initiation.

Interestingly, plant selection has been shown to impact the antioxidant ability of Cu-NMs. For example, Rehana et al. [211] synthesised nanoparticles using extracts of *Azadirachta indica*, *Hibiscus rosa-sinensis*, *Murraya koenigii*, *Moringa oleifera*, and *Tamarindus indica*, then tested their antioxidant capabilities with ABTS, DPPH, and hydrogen peroxide assays. *Tamarindus indica*-mediated nanoparticles were found to have the highest antioxidant activity, and *Moringa oleifera* the lowest, though still superior to CuO nanoparticles produced via a chemical method. Therefore, plant-mediated nanomaterials have much higher antioxidant ability as compared to chemical-mediated materials, and the plant used is an essential consideration for antioxidant purposes.

#### 4.1.2. Nano-Sensor

Plant-extract-mediated Cu-NMs have also been utilised in the preparation of nano-sensors. Cu nanomaterials, such as CuO nanoparticles, are suitable for nano-sensor production owing to their characteristic high electron-transfer rate, superior catalytic activity, large surface area, high glucose selectivity in heterogenous samples (such as blood or urine), chlorine poisoning resistance, and corrosion resistance. For example, Ag-CuO core-shell nanoparticles produced using *Ocimum tenuiflorum* extract have been used for non-enzymatic glucose sensing with a screen-printed electrode [174]. The synthesised electrode provided good glucose-sensing performance with a sensitivity of 3763.44 µAmM^−1^cm^−2^, linear range of 1 to 9.2 mM, detection limit of 0.006 mM (S/N = 3), and response time of less than 1 s. Moreover, the CuO-Ag core-shell-modified bio-nano-sensors demonstrated exceptional adhesion and structural strength along with great long-term stability for up to 60 days, exhibiting 99.2% of the initial value after one month with excellent repeatability and reproducibility.

The mechanism by which these nanoparticles sense glucose is based on electron transfer from the screen-printed electrode to the CuO nanoparticle core via the conduction band electrons of the Ag shell. This electron transfer occurs because the work function of CuO is bigger than that of Ag, and equalization of Fermi levels ensues after the materials come into electrical contact and the mobility of electrons is improved. The progression of current-induced charge carriers can boost electrocatalytic efficiency through a charge transfer mechanism; therefore, the Ag-CuO core-shell nanoparticles are electro-catalytically active and can induce electron-transfer reactions. The energy of a nanoparticle is dependent on the charge distribution within the energy levels of its component metal. Ultimately, the additional electrons can be discharged when glucose is introduced into the system as an electron acceptor.

This glucose oxidation mechanism can be summarised as: (1) deprotonation of glucose that causes oxidation, (2) isomerization and enediol formation, and, finally, (3) adsorption to the electrode surface, which leads to the oxidation of Cu(II)/Cu(III):(2)$$CuO+OH^{-}\;\to \;CuOOH+e^{-}$$
(3)$$\mathrm{CuOOH}+\mathrm{glucose}+{\;e}^{-}\to \;\mathrm{CuO}+{\mathrm{OH}}^{-}+\mathrm{gluconic}\;\mathrm{acid}$$

In the core-shell nanoparticle, Cu(II) was oxidised to Cu(III) and this catalysed glucose oxidation to produce gluconolactone, which was further oxidised to gluconic acid as presented in Equations (2) and (3).

#### 4.1.3. Anticancer

Lastly, plant extract-mediated Cu-NMs have been studied for their anticancer properties. Generally, this activity can be realised through multiple routes including ROS generation, antioxidant activity, cell cycle arrest, apoptosis, and autophagy [205,207]. One study produced CuO nanoparticles using extracts of *Azadirachta indica*, *Hibiscus rosa-sinensis*, *Murraya koenigii*, *Moringa oleifera*, and *Tamarindus indica* and used MTT assays to test their activity against four cancer cell lines, i.e., human breast, cervical, epithelioma, and lung cancer cells, along with one normal human dermal fibroblast (NHDF) cell line [211]. All CuO nanoparticles exhibited anticancer ability towards all cancer cell types in a dose-dependent manner: higher concentrations of CuO nanoparticles resulted in lower cancer cell viability. Interestingly, the type of plant utilised also affected anti-cancer ability, with *Tamarindus indica*-mediated CuO nanoparticles exhibiting greater cytotoxicity over the others; this indicates that the phytochemicals in the plant extract used for nanoparticle synthesis impact the resulting particles’ anti-cancer activity.

The toxicity of Cu-NMs is one of the limitations that hinder their application biomedically. However, it has been reported that plant-mediated Cu nanomaterials have less toxicity to normal human cell lines [205]. Therefore, such Cu nanomaterials may be more safely applied in biomedical applications. For example, CuO nanoparticles synthesised using extracts of *Azadirachta indica*, *Hibiscus rosa-sinensis*, *Murraya koenigii*, *Moringa oleifera*, and *Tamarindus indica* exhibited lower toxicity in NHDF cells, which suggests these to be promising anticancer agents for use in the pharmaceutical industry [211].

### 4.2. Environmental Remediation

The usage of Cu-NMs in environmental applications is mainly focused on the remediation of dyes and toxic compounds, with mechanisms primarily based on photocatalysis or catalysis.

The mechanism of photocatalysis by nanomaterials is as follows: when the nanomaterials are deposited into an aqueous sample containing compounds that are desired to be degraded, such as dye, and exposed to light, an interaction occurs in which a photogenerated electron is converted from the valence band (VB) to the conduction band (CB) in the nanomaterial. A hole in the VB then results, producing an electron (e^−^)-hole (h^+^) pair. The holes react with OH ions in the water molecules to yield OH radicals via oxidation, while the electrons react with dissolved O_2_ to generate O_2_ radicals via reduction. Those radicals are then responsible for the degradation of the dye into non-toxic degraded products [171,176]. Alshehri and Malik investigated the ability of *Origanum vulgare* extract-mediated Cu-Co-Ni trimetallic nanoparticles to photocatalyse the degradation of methylene blue [176]. They observed degradation efficiency of more than 50% and 92.67% after 50 and 100 min, respectively. The rate of degradation could be increased via increasing nanoparticle concentration, but after a certain threshold was surpassed, the photocatalytic efficiency could be enhanced no further due to the aggregation of the nanomaterials.

With regard to catalysis mechanisms, nanoparticles can catalyse reactions by borylation, clock reactions, oxidative coupling, A3 coupling, click chemistry, tandem and multicomponent reactions, C–H functionalization, cross-coupling, reduction and oxidation reactions, and other mixed reactions [212]. Successful catalysis via plant-mediated Cu-NMs has been achieved, such as when Suvarna et al. [173] studied the degradation of methyl green dye using bimetallic spherical Fe-Cu nanoparticles produced using *Cyclea peltata* extract, and achieved a degradation efficacy of 82% within 105 min. The Fe-Cu nanoparticles promoted hydrolysis and deprotonation reactions on the dye molecules, resulting in the demineralization of the dye molecules into simpler structures. In another example, Rosbero and Camacho utilised bimetallic (Ag and Cu) alloy nanoparticles produced via *Carica papaya* leaf extract to degrade the pesticide chlorpyrifos in water [172]. The degradation was observed for 24 h, and yielded the products 3,5,6-trichloropyridinol (TCP) and diethylthiophosphate (DETP), of which the former is less toxic than chlorpyrifos and not mutagenic.

## 5. Future Research Directions

Although plant-mediated Cu-NMs have numerous benefits and applications, they also have considerable potential yet to be discovered along with disadvantages that are unavoidable and need to be addressed to ensure realization of the applicability of these nanoparticles toward industrial production with wider applications. This section suggests areas of future research to increase the potential of Cu-NMs and propel the applicability of their production at a larger scale via eradicating current limitations; specifically, it discusses: (i) solutions by which to overcome limitations, (ii) potential new applications, and (iii) new research directions regarding Cu-NM synthesis.

### 5.1. Limitations and Solutions

This section illustrates the limitations of green Cu nanomaterial production and associated solutions. There are several that need attention in this respect, mainly with regard to biomass obtainability, the complexity of plant systems, the underlying synthesis process, nanomaterial quality, and low productivity.

Concerning bioresource accessibility, most research to date has focused on the use of local plant species that are not widely available throughout the globe; notably, variation in plant species and also geographical cultivation areas affect the phytochemicals within the plant extract produced [36]. Moreover, the inherent complexity of plants is another hindrance to the industrial production of plant-mediated Cu-NMs. That is, the phytochemicals within a plant are greatly affected by external factors such as abiotic environmental factors, cultivar, and mutagenesis [213,214,215]. These will cause batch-to-batch variation among raw materials, which might adversely affect the homogeneity and reproducibility of nanomaterial synthesis. On top of those considerations, another drawback to this method is the diversity of phytochemicals in a plant system. This can be addressed by applying molecular science techniques such as genetic engineering to maximize the most relevant phytochemicals in the target plant. Combining these techniques with plant-tissue culture methods such as cloning can allow the quality of a plant (target phytochemical composition and content) to be preserved and controlled and, thus, avoid batch-to-batch variation and mutagenic factors that might affect the phytochemical profile. Plant-tissue culture techniques can also minimize the time, cost, and labour force needed for the planting of bioresources and overcome geographical limitations [216].

At present, most research into plant-mediated nanoparticles is carried out in low quantities, and, hence, with low productivity. Although the quantity of nanoparticles required for characterization or application research purposes is not high, mass production via the green synthesis method is little-studied and needs to be researched in order to produce nanoparticles in a large quantity. Bioprocess methods can be used to produce and maximize specific phytochemicals for nanomaterial production and allow large-scale industrial production. With these solutions and more research, the global industrial production of uniform plant-mediated Cu nanomaterial products could be realised.

However, given the limited determinations of phytochemical profiles, it is not feasible to elucidate the mechanism of nanomaterial synthesis. This will affect the possibility of producing nanomaterials with good homogeneity in terms of size, shape, and crystal structure [71,75,79,84,217]. When it comes to investigating those phytochemicals that are responsible for stabilizing and reducing the ions within a precursor material, Fourier-transform infrared spectroscopy (FTIR) is the current technique of choice. This characterisation method mainly examines the functional groups that are deposited on the nanoparticle surface [71,79,84,177]. Although FTIR can identify the functional groups that act as stabilizing and reducing agents, it has difficulty determining which specific phytoconstituents of a complex plant extract they originated with. Further research employing other characterization methods such as liquid chromatography–mass spectrometry and nuclear magnetic resonance can be carried out to identified the chemical structures of the contributing phytochemicals [218,219].

### 5.2. Potential New Applications

At present, most Cu-NMs produced via plant-mediated methods are synthesised using leaf extracts. There is plenty of room for future research into the exploitation of other plant components (peelings, roots and rhizomes, fruits, flowers, and seeds) for the synthesis of Cu alloy, core shell, and nanoparticles. In addition, there remains a knowledge gap regarding the effect of method parameters on the morphology of the synthesised particles. From an application perspective, most uses of plant-mediated Cu nanomaterials are focused on biomedical and environmental remediation. However, there are more applications that have yet to be discovered. For example, in the biological sector, research into the use of plant-mediated Cu-NMs mainly concerns their antibacterial ability and lesser antifungal ability. The relatively lower antifungal performance is owed to fungal cells less readily adsorbing nanoparticles at low concentrations as compared to bacterial cells. In addition, the binding of nanoparticles to the bacterial surface blocks bacterial respiration, whereas for fungal or eukaryotic cells, respiration occurs in the mitochondrial membrane and so is less susceptible to direct inhibition by nanoparticles [31]. There has also been limited research on the antiviral and antiparasitic abilities of plant-mediated Cu nanoparticles; further investigation in this area may expand their antimicrobial capabilities. Finally, other noble-metal (Au and Ag) nanomaterials produced via plant-mediated synthesis have been employed in other applications such as electrochemistry, detection, surface-enhanced Raman scattering, phase transfer, transmetallation, and modified glassy carbon electrodes; therefore, research can be carried out to expand the usage of Cu-NMs to these applications [31].

### 5.3. New Research Directions for Synthesis

One area of future research for the application of Cu-NMs and their industrial-scale green production is leveraging machine learning. Specifically, machine-learning algorithms can be used in two respects, synthetic outcome prediction and experiment planning [220]. For the first, an algorithm mathematically learns the relationship between nanomaterial properties and experimental conditions, then predicts from an example synthetic parameters dataset and the results of past experiments the characteristics of the nanoparticles that will be produced. Meanwhile, experiment-planning algorithms aim to suggest the best reaction conditions for achieving desired nanomaterial properties [220]. This can aid in reducing the time and research effort required to obtain a desired outcome, such as the uniformity of the produced particles. Most machine-learning studies to date have concentrated on chemical-based nanomaterial synthesis; only a limited number of publications have concerned green synthesis; hence, there remains a large gap in this area [221,222,223,224]. Addressing this gap can help in making the process of Cu nanomaterial synthesis become less labour intensive, more cost effective, less time consuming, more productive, and able to yield higher quality nanoparticles, all of which are important from the industrial perspective.

## 6. Conclusions

This literature review focused on the green synthesis of Cu nanomaterials. Compared to chemical or physical synthesis methods, green synthesis and especially plant-mediated synthesis is more environment-friendly, less toxic, and safe throughout the whole production process. The production methodology was discussed with further focus on plant-mediated nanomaterial synthesis, including the plant extraction method and Cu-NM (pure metal, metal oxide, alloy, core shell, and nanoparticles) synthesis. Leaf-extract-mediated Cu nanomaterials comprise the majority produced to date, with few synthesised using other types of plant components. The review also considered the biological and environmental applications of plant-mediated Cu-NMs. With regard to biological applications, antiviral and antiparasitic activities have received less focus than antibacterial. There also remain many research gaps regarding the application of green synthesis Cu-NMs in other sectors. Finally, current limitations and solutions with potential future research targets were described. Biomass obtainability, complexity of plant systems, underlying synthesis process, nano-material quality, and low productivity are the future challenges that need to be addressed in order to further broaden the application of plant-mediated nanomaterial synthesis.

In short, plant-mediated nanomaterial synthesis is eco-friendly, has low toxicity, and avoids using hazardous chemicals. The process can be separated into two parts, the plant extraction and the nanomaterial production. Different plant extracts with different parameters can produce nanomaterials of different sizes and geometries. As such, plant source accessibility, diversity of phytochemicals in extracts, knowledge of the synthesis mechanism, and nanomaterial quality are the limitations that presently hinder the future industrial production and application of plant-mediated nanomaterials. More research is needed in areas of the biotechnological sector such as phytochemical profiling, molecular science, tissue culture, and bioprocesses to overcome these issues. Separately, machine learning can also be adopted as a new research topic to further improve the green synthesis of Cu-NMs with better industrial applicability. Once these problems and research directions are resolved and fulfilled, respectively, the potential of plant-mediated nanomaterial synthesis could be fully unleashed in myriad applications, providing processes and materials with better sustainability and friendliness toward the environment.

## Figures and Tables

**Figure 1 nanomaterials-12-03312-f001:**
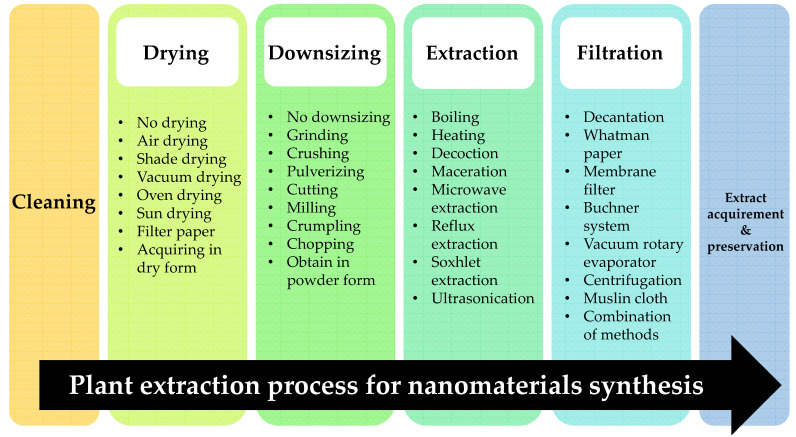
General steps in plant extraction.

**Figure 2 nanomaterials-12-03312-f002:**
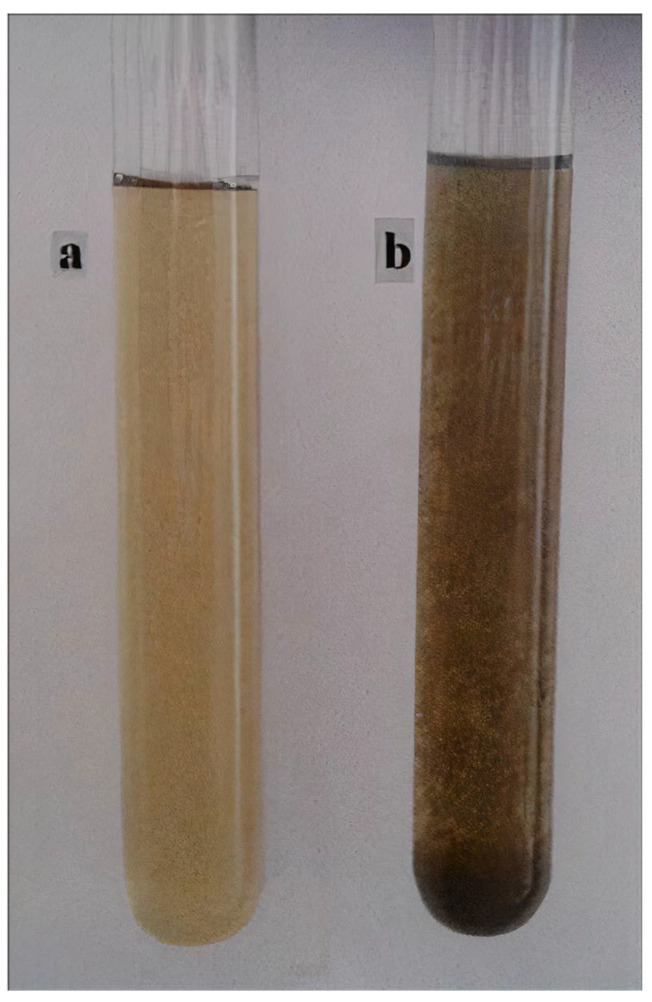
(**a**) *Thymus vulgaris* leaf extract and (**b**) solution after green synthesis of CuO nanoparticles. Adapted with permission from Ref. [85]. 2016, Elsevier.

**Figure 4 nanomaterials-12-03312-f004:**
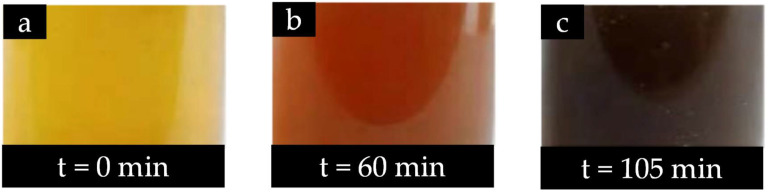
Colour change over time during the reaction between *Citrus reticulata* peel extract and CuSO_4_.5H_2_O at (**a**) 0 min, (**b**) 60 min and (**c**) 105 min. Adapted with permission from Ref. [186]. 2020, Elsevier.

**Figure 5 nanomaterials-12-03312-f005:**
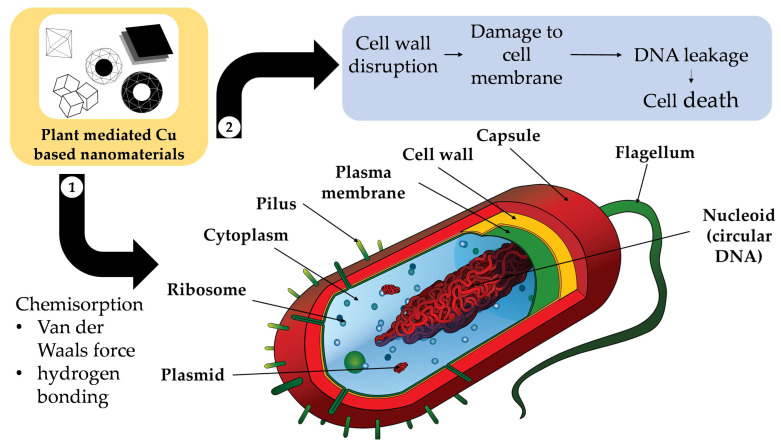
Diagram of the chemisorption-based mechanism of Cu-based nanomaterials’ antimicrobial activity.

**Figure 6 nanomaterials-12-03312-f006:**
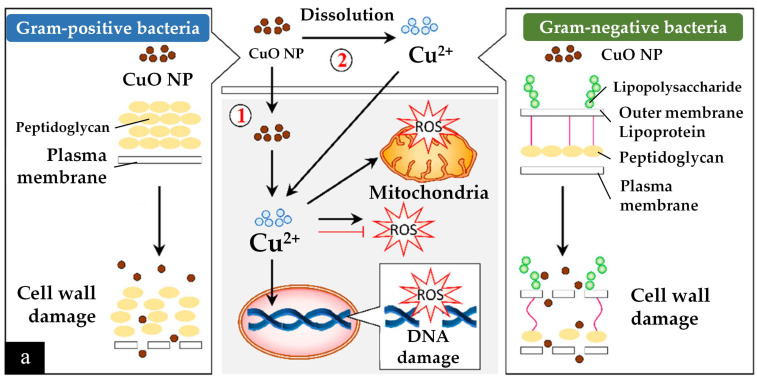
(**a**) Diagram of the respective mechanisms of CuO nanoparticle antibacterial activity in Gram-positive and Gram-negative bacteria and (**b**) diagram of the mechanism of CuO nanoparticle antifungal activity. Adapted with permission from Ref. [205]. 2020, MDPI.

**Table 1 nanomaterials-12-03312-t001:** Parameters and extraction method utilised for extraction of different plant components.

Species	Drying	Downsizing Method	Extraction Method	Temperature (°C)/Power	Time	Solvent	Reference
**Leaves**							
*Azadirachta indica*	Oven drying at 50 °C	-	Heating	60	20 min	DI-H_2_O	[65]
*Basella alba*	Shade drying at room temperature	Grinding and pulverizing	Boiling	60	20 min	DI-H_2_O	[66]
*Cacumen platycladi*	Acquired in dried form	Milling	Heating	30	4 h	DI-H_2_O	[67]
*Carica papaya*	Shade drying	Grinding	Boiling	60	30 min	DI-H_2_O	[68]
*Cymbopogon jwarancusa*	Shade drying at room temperature	Grinding	Boiling	Step 1: 100 Step 2: 37	Step 1: 30 min Step 2: overnight	Double DS-H_2_O	[69]
*Daphne mezereum*	Acquired in dried form	Acquired in cut form	Reflux extraction	-	15 min	DI-H_2_O	[70]
*Eclipta prostrata*	-	-	Boiling	80	30 min	Double DS-H_2_O	[71]
*Ixora brachypoda*	Air drying at room temperature	Cutting	Boiling	60	1 h	DI-H_2_O	[72]
*Iresine herbstii*	Drying	Pulverizing	Soxhlet extraction	-	12–16 h	Ethanol	[73]
*Jasminum sambac*	-	Cutting	Microwave irradiation	-	200 s	DS-H_2_O	[74]
*Magnolia kobus*	Drying at room temperature	Cutting	Boiling	-	5 min	DS-H_2_O	[75]
*Mentha* *aquatica*	Drying	Grinding	Ultrasonication	400 W	10 min (On/Off = 7 s/3 s)	DI-H_2_O	[76]
*Moringa oleifera*	Drying at room temperature	Grinding	Soxhlet extraction	35–45	10 h	Methanol	[77]
*Piper betle*	Shade drying at room temperature	Cutting	Boiling	-	5 min	Double-distilled deionised water	[78]
*Plantago asiatica*	Acquired in dried form	Acquired in powder form	Reflux extraction	80	30 min	Double DS-H_2_O	[79]
*Quercus coccifera*	Drying at room temperature	Grinding	Boiling	90	30 min	DS-H_2_O	[80]
*Ruellia tuberosa*	-	Chopping	Boiling	50	10 min	DI-H_2_O	[81]
*Solidago canadensis*	Drying at room temperature	Grinding	Heating	80	-	DS-H_2_O	[82]
*Syzygium cumini*	Oven drying at 60 °C	Crumpling	Boiling	100	35 min	DI-H_2_O	[83]
*Tabernaemontana divaricate*	-	Grinding	Boiling	-	10 min	DI-H_2_O	[84]
*Thymus vulgaris*	Acquired in dried form	Grinding	Reflux extraction	70	2 h	DS-H_2_O	[85]
*Tradescantia spathacea*	-	Chopping	Boiling	60	60 min	DI-H_2_O	[86]
*Camellia sinensis*	Acquired in dried form	-	Reflux extraction	-	40 min	DI-H_2_O	[87]
*Citrus limon*							
*Eucalyptus globulus*							
*Laurus nobilis*							
*Mentha* sp.							
*Quercus robur*							
*Rosmarinus officinalis*							
*Thimus mastichina*							
*Thimus vulgaris*							
*Thuja occidentalis*							
**Fruits**							
*Berberis vulgaris*	Acquired in dry form	Acquired in powder form	Heating	80	30 min	Double DS-H_2_O	[88]
*Capparis spinosa*	Oven drying (12 h) (383 K)	-	Boiling	-	30 min	Ethanol/ H_2_O (ratio-1:1)	[89]
*Citrus medica*	-	-	Squeezing to get juice	-	-	-	[51]
*Citrus sinensis*	-	Cutting	Squeezing to get juice	-	-	-	[90]
*Cleome viscosa*	-	-	Boiling	60	30 min	DS-H_2_O	[91]
*Couroupita guianensis*	Shade drying for 8–10 days	Chopping, grinding	Decoction	60	20 min	DS-H_2_O	[92]
*Crataegus pentagyna*	-	-	Maceration	-	-	Methanol	[93]
*Emblica officinalis*	-	Crushing	Boiling	-	10 min	Double DS-H_2_O	[94]
*Ficus carica*	Acquired in dry form	Chopping	Heating	100	1 h	Double DS-H_2_O	[95]
*Lycium barbarum*	-	-	Boiling	-	8 min	DI-H_2_O	[96]
*Piper longum*	Acquired in dry form	Acquired in powder form	Heating	70	30 min	30% methanolic solution	[97]
*Pouteria caimito*	Shade drying at room temperature	Cutting	Steeping	-	-	DS-H_2_O	[98]
*Sechium edule*	-	-	Heating	90 ± 2	12 h	DS-H_2_O	[99]
*Solanum mammosum*	Oven drying (25 °C)	Grinding	Mixing with solvent/maceration	-	1 h	DI-H_2_O	[100]
*Syzygium alternifolium*	Acquired in dry form	Acquired in powder form	Boiling	80	30 min	Milli-Q water	[101]
*Vaccinium macrocarpon*	Acquired in dry form	Grinding	Reflux extraction	90	45 min	DS-H_2_O	[102]
**Peelings**							
*Allium cepa*	Acquired in dry form	Cutting	Heating	90	30 min	DS-H_2_O	[103]
*Annona squamosa*	Air-drying	Grinding	Heating	60	30 min	Double DS-H_2_O	[104]
*Arachis hypogaea*	Oven drying method for 70 °C for 30 min	Peeling via oven drying method	Heating	70	30 min	Water	[105]
*Benincasa hispida*	-	-	Boiling	-	30 min	DS-H_2_O	[106]
*Carica papaya*	-	Acquired in small pieces	Heating	70–80	20 min	DI-H_2_O	[26]
*Citrus sinensis*	-	Smashing and grinding	Mixing	-	4 h	DI-H_2_O	[107]
*Garcinia mangostana*	Drying at ambient conditions; later with crude extract via oven drying	Grinding	Heating	80	1 h	Double DI-H_2_O	[108]
*Garcinia mangostana*	Oven drying at 40 °C	Grinding	Boiling	60	30 min	DS-H_2_O	[109]
*Myristica fragrans*	Acquired in dry form	Acquired in ground form	Boiling	100	1 h	DI-H_2_O	[110]
Orange peel	Drying by food drier for 12 h	Peeling and grinding	Stage 1: Maceration Stage 2: Heating	Stage 1: none Stage 2: 60	Stage 1: 3 h Stage 2: 60 min	DI-H_2_O	[111]
*Persea americana*	-	Milling	Maceration	-	24 h	DS-H_2_O	[112]
*Punica granatum*	Air-drying under shade	Chopping and grinding	Soxhlet extraction	55	30 min	DI-H_2_O	[113]
*Punica granatum*	Shade drying	-	Boiling	-	10 min	DS-H_2_O	[114]
*Punica granatum*	Oven drying 60 °C for 40 h	Acquired in powder form	Mixing	-	24 h	100% Ethanol	[115]
Tangerine	Shade drying (27 ± 2 °C)	Milling by electric mill and sieving	Heating	80	15 min	DS-H_2_O	[116]
*Citrus* *aurantifolia*	Drying via food dryer	Grinding	Stage 1: Maceration with solvent Stage 2: Heating	Stage 1: none Stage 2: 60	Stage 1: 3 h Stage 2: 60 min	DI-H_2_O	[117]
*Citrus paradisi*							
*Citrus sinensis*							
*Lycopersicon* *esculentum*							
**Flowers**							
*Aglaia elaeagnoidea*	Shade drying for 3 days	Grinding	Reflux extrication	-	10 min	DI-H_2_O	[118]
*Achillea wilhelmsii*	Air drying	Cutting	Boiling	-	10 min	Sterile DS-H_2_O	[119]
*Aloe vera*	Oven drying at 50 °C for 72 h	Cutting	Boiling	-	5 min	Double DS-H_2_O	[120]
*Avicennia marina*	-	Grinding	Boiling	-	5 min	DS-H_2_O	[121]
*Azadirachta indica*	Shade drying for a week	Crushing	Heating	80	1 h	DI-H_2_O	[122]
Calendula	Drying at room temperature	-	Heating	80	30 min	DI-H_2_O	[123]
*Gazania rigens*	Shade drying with oven drying	Cutting and grinding	-	-	3 h	Methanol	[124]
*Gnidia glauca*	Shade drying for 2 days at room temperature	Grinding	Boiling	-	5 min	DS-H_2_O	[125]
*Hibiscus sabdariffa*	Air drying under shade at room temperature		Soaking	Room temperature	2 h	DS-H_2_O	[126]
*Muntingia calabura*	-	-	Boiling via microwave oven	-	Boiling: 1 min * Process repeated at 1 h intervals for up to 6 h	DS-H_2_O	[127]
*Tagetes erecta*	-	Cutting	Boiling	-	10 min	Ultra-pure water	[128]
*Trifolium pratense*	Air drying for 5 days at room temperature	-	Heating	80	45 min	Double DS-H_2_O	[129]
**Roots and Rhizomes**							
*Berberis vulgaris*	Drying at ambient temperature for 2 days	Grinding	-	Room temperature	2 days	Sterile DS-H_2_O	[130]
*Bergenia ciliata*	Air drying at 25 °C	Acquired in powder form	Boiling	60	30 min	Milli-Q water	[131]
*Chromolaena odorata*	Sun drying At 22 °C ± 2 °C for 14 days	Crushing	Heating	85	2 h	DI-H_2_O	[132]
*Cibotium barometz*	Drying	Cutting and pulverizing	Boiling	100	30 min	DS-H_2_O	[133]
*Diospyros paniculata*	Air drying	Grinding	Soxhlet extraction	-	-	Methanol	[134]
Licorice	-	-	Heating	-	-	Ethanol and double-ionised water	[135]
*Morinda citrifolia*	Shade drying at room temperature	Grinding	Boiling	-	15 min	DS-H_2_O	[136]
*Nepeta leucophylla*	Shade drying for 30 days at room temperature (24–32 °C)	Grinding	Soxhlet extraction	Boiling point of methanol	8 h	Methanol	[137]
*Panax ginseng*	-	Cutting and grinding	Boiling	-	30 min	Sterile water	[138]
*Rheum palmatum*	Acquired in dry form	Acquired in powder form	Reflux extraction	80	45 min	Ethanol	[139]
*Rheum palmatum*	-	Acquired in powder form	Incubating/heating	40	24 h	Milli-Q DI-H_2_O	[140]
*Rhodiola rosea*	-	Grinding and screening via sieve	Boiling	100	30 min	DI-H_2_O	[141]
*Scutellaria baicalensis*	Acquired in dry form	Grinding	Autoclave heating	100	30 min	DS-H_2_O	[142]
*Zingiber officinale*	-	Grinding	Microwave		1 min	DI-H_2_O	[143]
*Zingiber officinale*	-	Cutting and pulverizing	Squeezing	-	-	-	[144]
**Seeds**							
*Bixa orellana*	Vacuum drying at 60 °C	Crushing	Steeping	In dark environment	24 h	Ethanol	[145]
*Caesalpinia bonducella*	-	Grinding	Sonication	-	30 min	DI-H_2_O	[146]
*Coffea arabica*	-	Grinding	Heating	85	25 min	DS-H_2_O	[147]
*Cucurbita pepo*	Shade air drying for 2 days	-	Heating	90	2 h	DS-H_2_O	[148]
*Eriobotrya japonica*	Oven drying at 50 °C for 24 h	Grinding	Heating	40	60 min	DI-H_2_O	[149]
*Persea americana*	Drying in dryer for 12 h	Grinding	-	Stage 1—room temperature Stage 2—65 ± 1	Stage 1: 60 min Stage 2: 60 min	DI-H_2_O	[150]
*Phoenix dactylifera*	-	Milling	Boiling	80	20 min	Sterile DS-H_2_O	[151]
*Phoenix sylvestris*	-	-	Steeping	45	12 h	Sterile double DI-H_2_O	[152]
Pomegranate	-	-	Crushing to get juice	-	-	DI-H_2_O	[153]
*Punica granatum*	Drying by pressing in filter paper	Grinding	Heating	80–85	10 min	Ultra-pure water	[154]
*Punica granatum*	-	Grinding	Mixing	-	2 h	Water	[155]
Quince	-	-	Heating	60	4 h	DS-H_2_O	[156]
*Salvia hispanica*	Drying	-	Heating	60	120 min	DS-H_2_O	[157]
*Tectona grandis*	Drying at room temperature for 3–4 days	Crushing	Boiling	80	15–20 min	Double DS-H_2_O	[158]
*Theobroma cacao*	Drying at room temperature for a week	Grinding	Maceration	-	A week	Methanol	[159]

**Table 2 nanomaterials-12-03312-t002:** Pros and cons of various plant extraction methods.

Extraction Methods		
	**Pros**	**Cons**
Boiling/heating/decoction	Water-soluble constituents can be extracted	Inefficient for light-/heat-sensitive compounds
Maceration	Simple Low cost and little experimental set-up Eco-friendly	Batch-to-batch variation potential Long extraction time
Microwave extraction	Fast extraction Less solvent needed Produce extract with high purity and phenolic yield Cost effective	High heat and energy loss during the extraction
Reflux extraction	Less solvent and extraction time required Good contact efficiency and mass transfer Simple and easy operation	Not suitable for thermolabile compounds
Soxhlet extraction	Displacement of transfer equilibrium between plant components and the solvent could be acquired High extraction temperature could be provided No filtration requirement after leaching	Large sample, extraction time, solvent requirements Excessive loss of heat energy
Ultrasonication	Less residence time of plant particles in the solvent Lower material and solvent requirements Fast extraction process	Energy intensive

**Table 3 nanomaterials-12-03312-t003:** Summary of plant-mediated Cu nanomaterial synthesis: plant extract type, key compounds, Cu precursors, synthesis time and temperature, reaction completion colour, and the Cu nanomaterial product, geometry, and size.

Plant	Cu Precursor	Synthesis Time	Synthesis Temperature (°C)	Key Compounds	Colour of the Product	Nanomaterials	Size (nm)	Geometry	Reference
**Leaves**									
*Agrimoniae herba*	K_2_PtCl_6_ CuSO_4_	4 h 8 h 16 h 24 h	65	Flavonoids	-	Core-shell Cu-core Pt-shell	30	Spherical	[170]
*Azadirachta indica*	Cu(NO_3_)_2_ AgNO_3_ ammonium molybdenate	Stage 1—26 h Stage 2—1 h	Stage 1—none Stage 2—500 (calcination)	-	-	CuO nanoparticles	-	Nanoflake	[171]
						Ag-CuO nanoparticles	-		
						Mo-CuO nanoparticles	-		
						Ag-Mo-CuO nanoparticles	12		
*Carica papaya*	CuSO_4_.5H_2_O	24 h	50–60	Flavonoids Phenolics	Green to blackish brown	CuO nanoparticles	<50	Spherical	[68]
*Carica papaya*	AgNO_3_ Cu(NO3)_2_	2 h	90	-	Light yellow green to olive green precipitate	Bimetallic Ag-Cu alloy	TEM-90-150 DLS-420.7	Tentacle-like	[172]
*Cyclea peltata*	FeSO_4_.7H_2_O CuSO_4_.5H_2_O	4 h	Room temperature	Carbohydrates Amino acids Alkaloids Flavonoids Saponins Gallotannins	Light yellow to green	Core-shell Cu-core Fe-shell	45–50	Spherical	[173]
*Eclipta prostrata*	Cu (CH_3_COO)_2_	24 h	Room temperature	-	-	Cu nanoparticles	28–45	Spherical, hexagonal, cubical	[71]
*Magnolia kobus*	CuSO_4_·5H_2_O	-	95	Terpenoids Reducing sugars	-	Cu nanoparticles	37–91	Spherical	[75]
			25				110		
			60				90		
*Ocimum tenuiflorum*	Cu (NO_3_)_2_ Ag (NO_3_)_2_	6 h	80	-	Brownish blue	Core-shell CuO-shell Ag-core	Ag core: 28–30 CuO shells: 6–10	Spherical	[174]
*Opuntia* *ficus-indica*	AgNO_3_ Cu (NO_3_)_2_	1 h	55	Ascorbic acid	Slight green shade	Core-shell Ag-coreCu-shell	10–20	Ellipsoidal	[175]
					Slight blue shade	Bimetallic Ag-Cu alloy		-	
*Origanum vulgare*	Cu(NO_3_)_2_·3H_2_O Ni(NO_3_)_2_·6H_2_O Co(NO_3_)_2_·6H_2_O	Until alteration of colour	40	Phenolic compounds Water-soluble glycosides Rosmarinic acid Water-soluble glycosides Caffeic acid Protocatechuic acid Glycoside protocatechuic acid Derivatives of rosmarinic acid 2-caffeoyloxy-3-[2-(4-hydroxy benzyl)-4,5-dihydroxy]phenylpropionic acid Flavonoids	Dark greenish-brown	Trimetallic Cu-Co-Nialloy	28.25	Nanoflake	[176]
*Pisonia grandis*	Zn(NO_3_)_2_.6H_2_O Mg(NO_3_)_2_.6H_2_O Cu(NO_3_)_2_.9H_2_O	Stage 1—4 h	Stage 1—80 Stage 2—450 (calcination)	Flavonoids	Green to brownish black	Zn-Mg-Cu oxide nanocomposites	50	Cubic	[177]
*Plantago asiatica*	CuCl_2_⋅_2_H_2_O	5 min	80	Polyphenolics	Dark	Cu nanoparticles	7–35	Spherical	[79]
*Tabernaemontana* *divaricate*	CuSO_4_	7–8 h	100	Enzymes Proteins	Brownish black	CuO nanoparticles	46 ± 4	Spherical	[84]
*Thymus vulgaris*	CuCl_2_.2H_2_O	5 min	60	Polyphenolics	Change from yellow to dark brown	CuO nanoparticles	<30	-	[85]
*Vitex negundo*	AgNO_3_ CuSO_4_	24 h	-	-	Green to Brown	Bimetallic Ag-Cu nanoparticles	60	Spherical	[178]
**Fruits**									
*Crataegus pentagyna*	FeCl_3_·6H_2_O FeCl_2_·4H_2_O Ag(NO_3_) Cu(NO_3_)_2_·3H_2_O tetra ethyl orthosilicate	-	Room temperature	-	-	Fe_3_O-SiO_2_- Cu_2_O-Ag nanocomposites	55–75	Spherical	[93]
*Piper retrofractum*	CuSO_4_·5H_2_O	60 min	60	Flavonoids Phenolic compounds Piperidine alkaloids Phenylpropanoids Amides	Dark green	Cu nanoparticles	2–10	Spherical	[179]
*Prunus nepalensis*	CuSO_4_	Overnight	Room temperature	-	Light green to brown and then to pink	Cu nanoparticles	35–50	Centred cubic	[180]
*Rosa canina*	Cu (CH_3_COO)_2_	1 h	100	-	Dark brown	CuO nanoparticles	15–25	Spherical	[181]
*Rubus glaucus*	Cu(NO_3_)_2_·3H_2_O	6 h	75–80	Flavonoids Phenolic compounds	-	CuO nanoparticles	45	Spherical	[182]
*Syzygium alternifolium*	CuSO_4_·5H_2_O	2 h	50	-	-	CuO nanoparticles	2–21	Spherical	[101]
*Ziziphus spina-christi*	CuSO_4_	-	80	Polyphenolic compounds	Green to reddish brown	Cu nanoparticles	5–20	Elongated spherical	[183]
**Peelings**									
*Carica papaya*	Cu(NO_3_)_2_·3H_2_O	Stage 1— none Stage 2—2 h	Stage 1—70–80 Stage 2—450 (calcination)	Phenolic compounds Flavonoids Catechins	Greenish-blue to green to dark green to black powder	CuO nanoparticles	85–140	Agglomerated spherical	[26]
Cavendish banana	Cu(NO_3_)_2_·3H_2_O	Stage 1— none Stage 2—2 h	Stage 1—Boiling Stage 2—400	-	Brown paste to black powder	CuO nanoparticles	50–85	Agglomerated spherical	[184]
*Citrus paradisi*(grapefruit)	Anhydrous CuSO_4_	Stage 1—20 min Stage 2—72 h	Stage 1—70 Stage 2—room temperature	-	Brown precipitate	Cu nanoparticles	56–59	Spherical	[185]
*Citrus reticulata*	CuSO_4_·5H_2_O	10 min	25 ± 2	-	Brown	Cu nanoparticles	54–72	Spherical	[186]
			30						
			40						
			50						
*Punica granatum*	CuSO_4_	Stage 1—10 min Stage 2—4 h	Stage 1—80 Stage 2—40	-	-	Cu nanoparticles	15–20	Spherical	[187]
**Flowers**									
*Acacia caesia*	AgNO_3_ CuNO_3_ ZnO nanoparticles	-	Stage 1—none Stage 2—400 (calcination)	-	-	Ag-Cu-ZnO nanocomposite	Ag -7 Cu- 12 ZnO-none	Spherical	[188]
	CuNO_3_ ZnO nanoparticles					Cu-ZnO nanocomposite	14		
*Aglaia elaeagnoidea*	Cu(NO_3_)_2_.3H_2_O	5 min	Room temperature	Phenolic compounds Proteins	Light brownish red to brick red	CuO nanoparticles	3–54	Spherical	[118]
*Aloe vera*	Cu (CH_3_COO)_2_	Stage 1—30 min Stage 2— overnight	Stage 1—50 Stage 2—room temperature	-	Light green to dark green	Cu nanoparticles	40	Spherical	[120]
*Azadirachta indica*	CuSO_4_·5H_2_O	1 h	80	Terpenoids Terpenes Flavonoids Alkaloids Carotenoids	Light blue to light green to dark yellow to brown precipitate	Cu nanoparticles	5	Spherical	[122]
*Bougainvillea sp.*	Cu (CH_3_COO)_2_	-	-	-	Blue to black-blue colour	CuO nanoparticles	12–20	Spherical	[188]
*Calendula sp.*	Fe_3_O_4_ nanoparticles Cu (NO_3_)_2_.3H_2_O	Stage 1—1 hStage 2—6 h	Room temperature	-	-	Cu-Fe_3_O_4_ nanocomposite	20–40	Globular	[123]
*Eichhornia crassipes*	CuSO_4_	48 h	Room temperature	Aromatic compounds like lawsone and phenol	Colourless to light red	Cu nanoparticles	12–15	Spherical	[189]
*Lantana camara*	Cu (CH_3_COO)_2_	Stage 1—10 min Stage 2—2 h	65	-	-	CuO nanoparticles	13–28	Spherical	[190]
**Roots and Rhizomes**									
*Asparagus adscendens*	CuSO_4_•5H_2_O	1 h	Room temperature	-	Pale yellow to sky blue	Cu nanoparticles	10–15	Spherical	[191]
*Asparagus racemosus*	Cu(NO_3_)_2_.3H_2_O	8 h	60	Phenolic compounds	-	CuO nanoparticles	Diameter: 50–100 Length: 400–500	Rod-like	[192]
*Corallocarbus epigaeus*	CuSO_4_	12 h	80–100	-	Deep blue to colourless and then to brick red and dark red	Cu nanoparticles	65–80	Spherical	[193]
*Polyalthia longifolia*	CuSO_4_	30 min with stirring and 24 h storage	-	Phenolic compounds Flavonoids	Dark green colour	Cu, CuO_2_, Cu_2_O, and CuO nanoparticles	30	Spherical	[194]
*Rheum emodi*	AgNO_3_ Cu (CH_3_COO)_2_	3 h	90	Physcion Chrysophanol Aloe-emodin Emodin Chrysophanol glycoside	Light brown to black	Bimetallic Ag-Cu nanoparticles	40–50	Pseudo-spherical	[195]
	Cu (CH_3_COO)_2_	4 h	90		Blue to brown	Cu nanoparticles	-	-	
*Senna didymobotrya*	CuSO_4_·5H2O	-	40	Alizarin Quercetin	-	Cu nanoparticles	5.55–63.60	Spherical	[196]
			60						
			80						
*Zingiber officinalis*	Copper sulphate	-	Room temperature	-	Straw yellow to sea green	Cu nanoparticles	Around 20–100	Spherical	[197]
*Curcuma longa*									
**Seeds**									
*Caesalpinia bonducella*	Cu(NO_3_)_2_·3H_2_O	Stage 1—5 hStage 2—2 h	-	-	Stage 1—blue-coloured solution turned green Stage 2—dark brown precipitate	CuO nanoparticles	-	Rice-grain-shaped	[146]
*Carum carvi*	-	-	-	-	-	Cu nanoparticles	37	Spherical	[198]
						Fe_3_O_4-_Cu nanocomposite	62	Spherical	
*Koelreuteria apiculata*	CuCl_2_.2H_2_O	24 h	-	-	Precipitate formation	Cu nanoparticles	20	Spherical	[199]
*Persea americana*	CuSO_4_	6–7 h	45–50	Flavonoids Phenolic compounds	Brownish black	Cu nanoparticles	42–90	Spherical	[200]
*Punica granatum*	CuCl_2_·2H_2_O	Stage 1—10 min Stage 2—1–2 h Stage 3—4–6 h	Stage 1—60–70 Stage 2—60 Stage 3—room temperature	Alkaloids Flavonoids Polyphenols	Stage 2—dull bluish brown colour Changed to dark green	Cu nanoparticles	40–80	Spherical	[154]
*Silybum marianum*	FeCl_3_·6H_2_O CuCl_2_.2H_2_O	5 h	60	Flavonoids Phenolic compounds	Dark solution and forming of precipitate	Cu-Fe_3_O_4_ nanoparticles	8.5–60	Spherical	[201]
*Theobroma cacao*	PdCl_2_ CuCl_2_·2H_2_O	2 h	50	Flavonol antioxidants such as epicatechin, catechin	-	Pd-CuO nanoparticles	40	-	[202]
*Triticum aestivum*	CuSO_4_·5H_2_O	Stage 1—1 h Stage 2—10 min Stage 3—20 min	Stage 1—room temperature (25) Stage 2—sonication Stage 3—70	Starch	Dark blue to dark brown	CuO nanoparticles	21–42	Spherical	[203]

## Data Availability

Not applicable.

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
