# Peer review of "Biogenic Synthesis of Copper-Based Nanomaterials Using Plant Extracts and Their Applications: Current and Future Directions"

_nanomaterials, 2022, doi:10.3390/nano12193312_

Round 1
Reviewer 1 Report
The authors achieve, in an outstanding manner, the preparation of a well-written and well-documented review article. The topic relates to the preparation of copper-based nanostructures using plant extracts, and most importantly, they give the proper space to mention and explain some of the most relevant applications in different research fields. Another strong point of the manuscript is the detailed explanation of some protocols to obtain extracts; in this sense, this work would reach the interest of a broad audience, those who want to obtain Cu structures, plant extracts, or both, separately or simultaneously. Finally, the amount of information provided by the manuscript is extensive but well-summarized; thus, this work could be a good piece of information to check frequently. Therefore, I recommend the publication of the manuscript in its current form.
Author Response
Thank you for the positive comments and recommendation. No revision was made as no amendment needed.
Reviewer 2 Report
The authors review the use of plant extracts to synthesize copper-based nanomaterials. They first start by reviewing the traditional methods, indicating their advantages and problems and then review all green methods that have been used or are currently being used (bacteria, fungi, algae, etc) for the fabrication of nanomaterials. They review then all methods and their stages that are used in the synthesis of nanomaterials using plant extracts.
The review is well made and all relevant references of the field seem to be included. The article is well written and the figures and tables are, in general, clear in and easy to follow. I think it might be considered for publication in nanomaterials after the following issues have been addressed:
- It is difficult to read and follow the article due to a lack of subsections in some parts. For instance, section 3 would be clearer with a series of subsections for each of the synthesis steps.
- Some figures are not easy to read, their letters are rather small and difficult to understand (see e.g. Fig. 6). It would also be good to include some more figures, specially in the first pages of the manuscript.
- Although this is a review on copper related materials, there are many references to other materials that are used, specially in the first stages of the article, to comment on some methods or steps of the fabrication. It would be good to resort mainly to copper systems and, in case that is not possible, comment if that particular study can be applied to copper systems.
Reviewer 3 Report
The authors have reviewed the uses of plant extracts to prepare Cu-based nanomaterials and their robust applications. The mechanism of plant-mediated Cu-based nanomaterials in biomedical and environmental applications have been also summarized in this review. This review has provided readers with comprehensive information, including guidance, and future research directions of plant extraction, plant-mediated synthesis of Cu-based nanomaterials, the applications of plant-mediated Cu-based nanomaterials in biomedical and environmental remediation, and future research directions in this area. Overall, this review can inspire more material synthetic ideas of plant-mediated synthesis of Cu-based nanomaterials. Therefore, I would like to recommend this review to publish in Nanomaterials. Below are some comments for the authors.
1. This review would be more impressive if the authors could provide the future challenge of plant-mediated synthesis of Cu nanomaterials in the section of “Conclusions”.
2. For the introduction “Cu is an element that has drawn significant attention from researchers in nanotechnology, specifically in the nanomaterial sector”, more references could be cited to broaden the introduction.
https://doi.org/10.1016/j.jcis.2021.10.019
